# A Frustratingly Easy Plug-and-Play Detection-and-Reasoning Module for Chinese Spelling Check

**Haojing Huang**[1*]**, Jingheng Ye**[1*]**, Qingyu Zhou**[2†]**, Yinghui Li**[1]
**Yangning Li**[1,3]**, Feng Zhou**[2]**, Hai-Tao Zheng**[1,3,†]

[1]Tsinghua Shenzhen International Graduate School, Tsinghua University
[2]OPPO Research Institute, [3]Peng Cheng Laboratory
{hhj23, yejh22, liyinghu20}@mails.tsinghua.edu.cn
qyzhgm@gmail.com, zheng.haitao@sz.tsinghua.edu.cn

## Abstract

In recent years, Chinese Spelling Check (CSC) has been greatly improved by designing task-specific pre-training methods or introducing auxiliary tasks, which mostly solve this task in an end-to-end fashion. In this paper, we propose to decompose the CSC workflow into *detection*, *reasoning*, and *searching* subtasks so that the rich external knowledge about the Chinese language can be leveraged more directly and efficiently. Specifically, we design a plug-and-play detection-and-reasoning module that is compatible with existing SOTA non-autoregressive CSC models to further boost their performance. We find that the detection-and-reasoning module trained for one model can also benefit other models. We also study the primary interpretability provided by the task decomposition. Extensive experiments [1] and detailed analyses demonstrate the effectiveness and competitiveness of the proposed module.

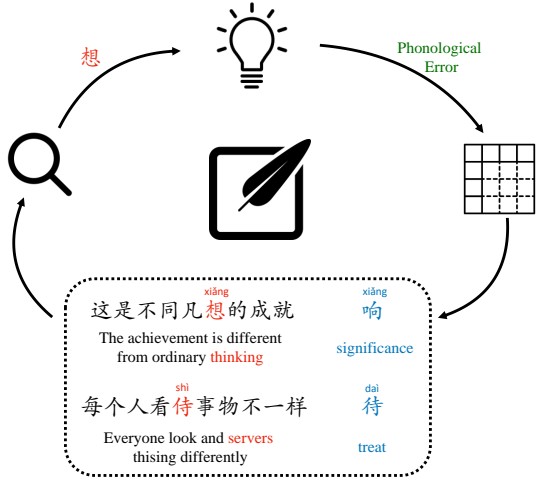

Figure 1: Examples of the CSC task. The first example is a phonological error, and the second is a morphological error. We mark the wrong character in red, the right character in blue, and also the error type of the character in green.

## 1 Introduction

Spelling Check aims to detect and correct spelling errors contained in the text (Wu et al., 2013a). It benefits various applications, such as search engine (Gao et al., 2010; Martins and Silva, 2004), and educational scenarios (Afli et al., 2016; Dong and Zhang, 2016; Huang et al., 2016, 2018; Zhang et al., 2022b). Particularly, in this paper, we consider spelling check for the Chinese language, i.e., Chinese Spelling Check (CSC), because of the challenge brought by the peculiarities of Chinese to the task (Li et al., 2021b). Suffering from many confusing characters, Chinese spelling errors are mainly caused by phonologically and visually similar characters (Liu et al., 2010). As shown in Figure 1, "侍(shì)" is the visual error character of "待(dài)" because they have almost the same strokes.

Recently, with the rapid development of Pre-trained Language Models (PLMs) (Dong et al., 2023; Li et al., 2023c; Ma et al., 2023), methods based on PLMs have gradually become the mainstream of CSC research (Li and Shi, 2021; Ji et al., 2021; Wang et al., 2021; Li et al., 2022f; Cheng et al., 2023a), most of which are non-autoregressive models making predictions for each input Chinese character. Previous works can be roughly divided into two categories: (1) Design tailored pre-training objectives to guide the model to learn the deep semantic information (Liu et al., 2021; Zhang et al., 2021b; Li, 2022; Li et al., 2023a); (2) Perform different fine-tuning processes like adding auxiliary tasks to fine-tune the models (Zhu et al., 2022; Liu et al., 2022b).

Despite the remarkable success achieved by previous works, existing models seem to have some shortcomings in incorporating external expert knowledge, such as the confusion set, which contains a set of similar character pairs (Liu et al.,

---

*  indicates equal contribution.
†  Corresponding authors: Hai-Tao Zheng, Qingyu Zhou.
[1]The source codes are available at https://github.com/THUKElab/DR-CSC.

2010; Wang et al., 2019). Cheng et al. (2020) makes an attempt that replaces the classifier's weight according to the confusion set using graph neural networks. Li et al. (2022e) further proposes a neural model that learns knowledge from dictionary definitions. However, these approaches implicitly learn external knowledge, thus lacking the desired level of interpretability and efficiency. We argue that the CSC task can be decomposed into three subtasks, including **Detection**, **Reasoning**, and **Searching**, aiming to incorporate the external knowledge more naturally and provide more interpretability. These subtasks answer the following questions respectively, "Which character is misspelled?", "Why did this error occur?" and "How to correct it?". By addressing these questions, the model gains access to error position, misspelling type, and external knowledge from three subtasks, respectively. As a result, a deeper understanding of the underlying processes involved in correcting misspelled characters can be achieved.

To this end, we propose a frustratingly easy plug-and-play **D**etection-and-**R**easoning module for **C**hinese **S**pelling **C**heck (**DR-CSC**) to incorporate external expert knowledge naturally and effectively into existing non-autoregressive (Cheng et al., 2023d) CSC models. The proposed method is designed to be compatible with a wide range of non-autoregressive CSC models, such as Soft-Masked BERT (Zhang et al., 2020), MacBERT (Cui et al., 2020), and SCOPE (Li et al., 2022b). We decompose the CSC task into three subtasks from easy to hard: (1) *Detection*: We model the process of detecting misspellings from a sentence through the sequence labeling task, that is, predicting for each character in the sentence whether it is correct or wrong. (2) *Reasoning*: For the wrong characters predicted in the detection step, we use the representations of CSC models to further enable the model to consider the attribution of spelling errors by classifying them to know whether they are errors caused by phonetics or vision. (3) *Searching*: After determining the spelling error type, we search within its corresponding phonological or visual confusion set to determine the correct character as the correction result. The proposed DR-CSC module performs multi-task learning on these three subtasks. By doing so, it naturally incorporates the confusion set information, and effectively narrows the search space of candidate characters. As a result, this module greatly helps to improve the

performance of CSC models.

In summary, the contributions of this work are in three folds:

- We design the DR-CSC module, which guides the model to correct Chinese spelling errors by incorporating the confusion set information. And it is also compatible with non-autoregressive CSC models.

- We enhance the interpretability of CSC models by explicitly incorporating external knowledge through the decomposition of the CSC task into three subtasks.

- We conduct extensive experiments on widely used public datasets and achieve state-of-the-art performance. Detailed analyses show the effectiveness of the proposed method.

## 2 Related Work

CSC is a fundamental language processing task, and it is also an important subtask of Chinese Text Correction (Ma et al., 2022; Li et al., 2023b; Ye et al., 2023, 2022; Zhang et al., 2023). Around the pre-training and fine-tuning of PLMs (Liu et al., 2022a; Li et al., 2022d,c; Cheng et al., 2023b; Zhang et al., 2022a; Cheng et al., 2023c), researchers have done many efforts to improve the performance of CSC models:

**CSC-targeted Pre-training Tasks** Researchers design different pre-training strategies to obtain PLMs that are more suitable for CSC. PLOME (Liu et al., 2021) proposes to pre-train models with their designed particular character masking strategy guided by the confusion set and apply task-specific pre-training to enhance the CSC models. Spell-BERT (Ji et al., 2021), DCN (Wang et al., 2021), and MLM-phonetics (Zhang et al., 2021a) leverage the phonetic information to improve the adaptation of the pre-training process to the CSC task. For more recent research, WSpeller (Li et al., 2022a) trains a word-masked language model to utilize the segment-level information which provides appropriate word boundaries.

**CSC-targeted Fine-Tuning Processes** Many studies focus on the fine-tuning stage of PLMs to obtain various additional knowledge to improve model performance. REALISE (Xu et al., 2021) and PHMOSpell (Huang et al., 2021) focus on the positive impact of multimodal knowledge on

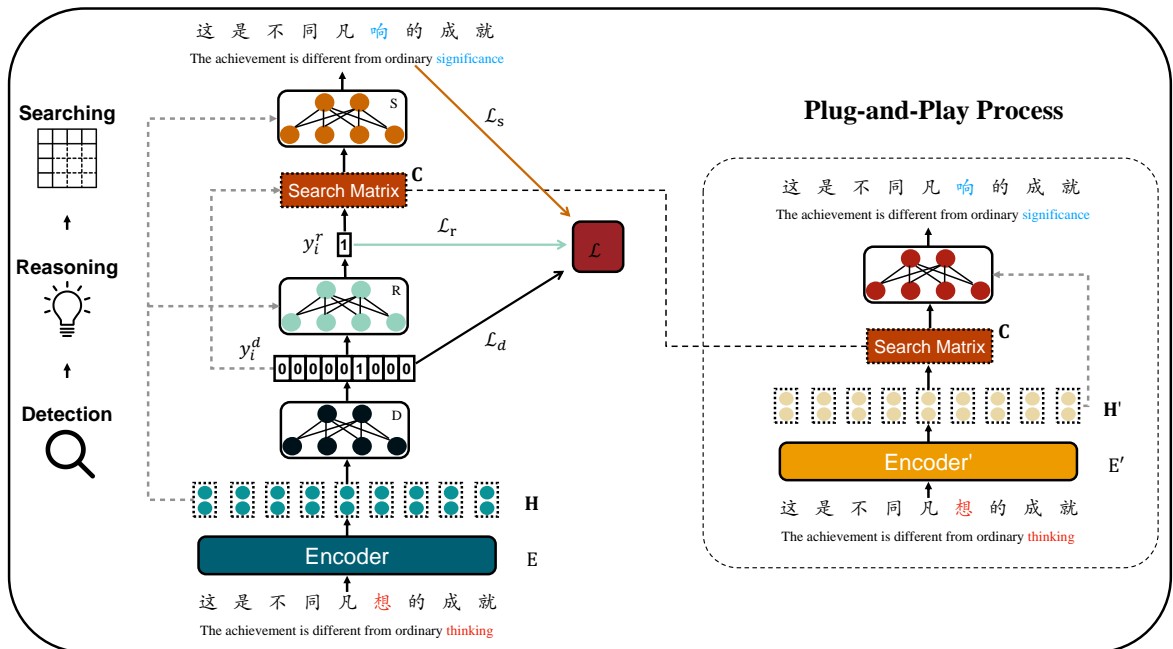

Figure 2: Overview of the proposed DR-CSC. The Detection task is to detect the error character. The Reasoning task is to determine the error type of the typo. The Searching task is to correct the sentence in the search matrix. The right part shows how to plug and play DR-CSC module from models to enhance another model.

the CSC models. ECOPO (Li et al., 2022f) and EGCM (Sun et al., 2022) propose error-driven methods to guide models to avoid over-correction and make predictions that are most suitable for the CSC scenario. Besides, LEAD (Li et al., 2022e) is a heterogeneous fine-tuning framework, which for the first time introduces definition knowledge to enhance the CSC model. SCOPE (Li et al., 2022b) proposes the auxiliary task of Chinese pronunciation prediction to achieve better CSC performance. MDCSpell (Zhu et al., 2022) proposes to fuse the detection network output into the correction network to reduce the impact of typos.

Although PLMs have achieved great success in the CSC task, previous studies still lack a degree of interpretability. We argue that the CSC task can be decomposed into three subtasks to introduce external knowledge naturally and provide more possibility of interpretability. The proposed module is compatible with existing non-autoregressive CSC models to get enhancement.

## 3 Methodology

As shown in Figure 2, DR-CSC consists of an encoder (i.e., E) and three progressive subtasks. DR-CSC is a module that can be combined with various existing non-autoregressive CSC models or proper PLMs.

### 3.1 Detection

Given the input sentence $\mathbf{X} = \{x_1, x_2, ..., x_T\}$ of length $T$, we utilize the PLMs or CSC models as encoder E to get the representations:

$$\mathbf{H} = \mathrm{E}(\mathbf{X}) = \{h_1, h_2, ..., h_T\}, \quad (1)$$

where $h_i \in \mathbb{R}^{hidden}$, $hidden$ is the hidden state size of E. The detection task is to detect the wrong characters from the sentence, that is, to predict whether each character in the sentence is correct or wrong. Therefore, we compute the binary classification probability of the $i$-th character in the sentence $\mathbf{X}$ as:

$$p_i^d = \mathrm{softmax}(W_D h_i + b_D), p_i^d \in \mathbb{R}^2, \quad (2)$$

where $W_D \in \mathbb{R}^{2 \times hidden}$ and $b_D \in \mathbb{R}^2$ are learnable parameters. Note that $p_i^d$ is a probability vector with 2-dimension, based on which we obtain the detection predicted result:

$$y_i^d = \arg\max(p_i^d), \quad (3)$$

where $y_i^d$ belongs to $\{0, 1\}$, $y_i^d = 0$ means that the character is correct, and $y_i^d = 1$ means that the character is wrong.

For the training of the detection task, we use the cross-entropy as its learning objective:

$$\mathcal{L}_d = -\sum_{i=1}^{T} g_i^d \log p_i^d, \quad (4)$$

where $g_i^d$ is the training label of the $i-$th character in the detection task. It is worth mentioning that during the training process, $g_i^d$ can be obtained by comparing the original character with its corresponding golden character.

## 3.2 Reasoning

After the detection task, we know which characters in the sentence are misspellings, then the reasoning task is to predict whether the wrong character is caused by phonetics or vision. It is not difficult to know that this is also actually a binary classification process for characters. Therefore, similar to the detection task, we calculate the reasoning prediction probability of the $i-$th character and get the reasoning predicted result as follows:

$$p_i^r = \mathrm{softmax}(W_R h_i + b_R), p_i^r \in \mathbb{R}^2, \quad (5)$$

$$y_i^r = \arg\max(p_i^r), \quad (6)$$

where $W_R \in \mathbb{R}^{2 \times hidden}$ and $b_R \in \mathbb{R}^2$ are learnable parameters. When $y_i^r = 1$, it means this wrong character is a phonological error. And $y_i^r = 0$ represents the morphological error. We also utilize the cross-entropy loss to train the reasoning task:

$$\mathcal{L}_r = -\sum_{i=1}^{T} y_i^d g_i^r \log p_i^r. \quad (7)$$

Note that we use $y_i^d$ of Equation (3) to ensure that only the wrong characters detected by the detection task participate in the calculation of $\mathcal{L}_r$. The reasoning label $g_i^r$ of a character is obtained by considering its golden character. If the golden character is in the phonological confusion set of the original character, its label is "phonological error". If the case is visual confusion set, the label is "morphological error". Specifically, we construct the phonological/visual confusion set of a character by comparing the Pinyin/strokes [2] between characters. The phonological confusion set of a character contains all characters similar to its pinyin, and the visual confusion set is all characters similar to its strokes. We also utilize the confusion set provided in SIGHAN13[3]. It contains Similar Pronunciation confusion set and Similar Shape confusion set. Previous research (Liu et al., 2010) indicates that 83% of errors are phonological errors and 48%

are morphological errors. This is because a large number of characters exhibit errors in both phonological and morphological aspects simultaneously, so it is reasonable for us to prioritize this kind of wrong character as "phonological error". During the construction of the confusion set, we ensured that the corresponding characters were included. This approach helps alleviate errors caused by the misidentification of non-erroneous characters as errors. Furthermore, the sets of phonetically similar characters and visually similar characters are not mutually exclusive; they exhibit a certain degree of intersection. This characteristic further mitigates the impact of error accumulation.

## 3.3 Searching

The searching task is to find the correct answers to the spelling errors. Specifically, for each character in $\mathbf{X}$, we predict the probability that it should be a character in the vocabulary of the PLMs:

$$p_i^s = \mathrm{softmax}(W_S h_i + b_S), p_i^s \in \mathbb{R}^{vocab}, \quad (8)$$

$$\mathbf{P}^s = \{p_1^s, p_2^s, ..., p_T^s\}, \mathbf{P}^s \in \mathbb{R}^{T \times vocab}, \quad (9)$$

where $W_S \in \mathbb{R}^{vocab \times hidden}$ and $b_S \in \mathbb{R}^{vocab}$ are trainable parameters, $vocab$ is size of PLMs' vocabulary.

The main innovation of our designed searching task is that we utilize the phonological/visual confusion set to enhance the prediction of error correction results. Thanks to the error position and error type information obtained in the detection and reasoning tasks, we can construct a more refined search matrix based on the phonological/visual confusion sets to reduce the search space:

$$c_i = \begin{cases} \vec{pc}[x_i], & y_i^d = 1 \ \& \ y_i^r = 1 \\ \vec{vc}[x_i], & y_i^d = 1 \ \& \ y_i^r = 0, \\ \vec{1}_{vocab}, & otherwise \end{cases} \quad (10)$$

$$\mathbf{C} = \{c_1, c_2, ..., c_T\}, \\ c_i \in \mathbb{R}^{vocab}, \mathbf{C} \in \mathbb{R}^{T \times vocab}, \quad (11)$$

where $\vec{pc}$ and $\vec{vc}$ are vectors whose dimensions are $vocab$ and whose elements are 0 or 1. And $\vec{pc}[x_i]$ means that the elements at the corresponding position of $x_i$'s phonetically similar characters in the vector are set to 1, and the others are 0. The elements of $\vec{vc}[x_i]$ are also set based on the visually similar characters of $x_i$.

---

[2]We utilize the cnchar toolkit (https://github.com/theajack/cnchar) to get Pinyin/strokes information of Chinese characters.

[3]http://ir.itc.ntnu.edu.tw/lre/sighan7csc.html

We augment the correction probabilities based on $\mathbf{C}$:

$$\mathbf{P} = \mathbf{P}^s \odot \mathbf{C} = \{p_1, p_2, ..., p_T\}, \quad (12)$$

where $\odot$ means the multiplication of elements in the corresponding positions of the two matrices. Through this operation, for a character that has been determined to be a phonological error, the probability that it is predicted to be a character that is not similar to its phonetic will be set to 0. The same is true for the visually similar case. Therefore, we enhance the probability representation of more suitable candidate characters through the search matrix in the searching task and also narrow the search space of candidate characters. We also use the cross-entropy loss in the searching task:

$$\mathcal{L}_s = -\sum_{i=1}^{T} g_i \log p_i. \quad (13)$$

Note that the label $g_i$ is equivalent to the overall golden label, i.e., the correct character corresponding to the wrong character.

Finally, benefiting from multi-task learning, we tackle the three progressive tasks at once in our DR-CSC framework:

$$\mathcal{L} = \alpha \cdot \mathcal{L}_d + \beta \cdot \mathcal{L}_r + \gamma \cdot \mathcal{L}_s, \quad (14)$$

where $\alpha$, $\beta$, and $\gamma$ are hyper-parameters.

### 3.4 Plug-and-Play Process

As depicted in Figure 2, the plug-and-play integration of the trained encoder E and detection-and-reasoning module (D-R Module) with a new CSC model or PLM E$'$ can be achieved without retraining of the D-R Module. This is accomplished by feeding the text to both encoders, E and E$'$:

$$\mathbf{H} = \mathrm{E}(\mathbf{X}), \quad \mathbf{H}' = \mathrm{E}'(\mathbf{X}). \quad (15)$$

$\mathbf{H}$ can then be utilized as input for the D-R Module, yielding the results of detection and reasoning sub-tasks for constructing a search matrix $\mathbf{C}$ according to Equations (2)(3)(5)(6)(10). $\mathbf{H}'$ can be fed into the output layer to obtain the correction probabilities $\mathbf{P}'$ of the new CSC model or PLM:

$$\mathbf{C} = \text{D-R Module}(\mathbf{H}), \quad (16)$$

$$\mathbf{P}' = \text{Output Layer}(\mathbf{H}'). \quad (17)$$

Subsequently, we can augment the correction probabilities $\mathbf{P}'$, by referring to Equation (12).

## 4 Experiments

### 4.1 Experimental Setup

#### 4.1.1 Datasets

To ensure fairness, we use the same training data as our baselines, i.e., SIGHAN13 (Wu et al., 2013b), SIGHAN14 (Yu et al., 2014), SIGHAN15 (Tseng et al., 2015) and Wang271K (Wang et al., 2018). The test data is also widely used in previous CSC task (Wang et al., 2019; Cheng et al., 2020; Li et al., 2022a), i.e, SIGHAN13/14/15 test datasets.

#### 4.1.2 Baselines Methods

To evaluate the performance of DR-CSC, we select some advanced and strong CSC models as our baselines: **SoftMasked-BERT** (Zhang et al., 2020) is consist of Detection Network and Correction Network. **MacBERT** (Cui et al., 2020) is an enhanced BERT with novel MLM as the correction pre-training task. **Two-Ways** (Li et al., 2021a) utilizes the weak spots of the model to generate pseudo-training data. **MLM-phonetics** (Zhang et al., 2021a) is an enhanced ERNIE (Sun et al., 2020) which contains additional phonetic information. **REALISE** (Xu et al., 2021) is a multi-modal CSC model which leverages semantic, phonetic, and graphic knowledge. **LEAD** (Li et al., 2022e) learns heterogeneous knowledge from the dictionary, especially the knowledge of definition. **SCOPE** (Li et al., 2022b) improve the CSC performance with the help of an auxiliary Chinese pronunciation prediction task. It is the previous state-of-the-art method for the SIGHAN datasets.

#### 4.1.3 Evaluation Metrics

Because CSC aims to detect and correct spelling errors, the CSC studies all report the detection and correction performance. For the correction level, the model must not only detect but also correct all wrong characters. More specifically, we report the sentence-level metrics, including Precision, Recall, and F1 score, which are commonly used in previous works (Xu et al., 2021; Li et al., 2022b). Note that sentence-level is more challenging than character-level because some sentences may contain multiple wrong characters.

#### 4.1.4 Implementation Details

In the experiments, we use Pytorch to implement the proposed DR-CSC. The implementations of SoftMasked-BERT + DR-CSC, MacBERT + DR-CSC and SCOPE + DR-CSC are following

| Dataset | Model | Detection | | | Correction | | |
|---|---|---|---|---|---|---|---|
| | | **P** | **R** | **F** | **P** | **R** | **F** |
| **SIGHAN13** | MLM-phonetics (Zhang et al., 2021a) | 82.0 | 78.3 | 80.1 | 79.5 | 77.0 | 78.2 |
| | REALISE (Xu et al., 2021) | **88.6** | 82.5 | 85.4 | 87.2 | 81.2 | 84.1 |
| | Two-Ways (Li et al., 2021a) | - | - | 84.9 | - | - | 84.4 |
| | LEAD (Li et al., 2022e) | 88.3 | 83.4 | 85.8 | 87.2 | 82.4 | 84.7 |
| | SoftMasked-BERT (Zhang et al., 2020)† | 81.1 | 75.7 | 78.3 | 75.1 | 70.1 | 72.5 |
| | SoftMasked-BERT + DR-CSC | 80.2 | 77.3$^\uparrow$ | 78.7$^\uparrow$ | 77.9$^\uparrow$ | 75.1$^\uparrow$ | 76.5$^\uparrow$ |
| | MacBERT (Cui et al., 2020)† | 84.6 | 79.9 | 82.2 | 81.0 | 76.5 | 78.7 |
| | MacBERT + DR-CSC | 87.8$^\uparrow$ | 81.6$^\uparrow$ | 84.6$^\uparrow$ | 86.7$^\uparrow$ | 80.5$^\uparrow$ | 83.5$^\uparrow$ |
| | SCOPE (Li et al., 2022b) | 87.4 | 83.4 | 85.4 | 86.3 | 82.4 | 84.3 |
| | SCOPE + DR-CSC | 88.5$^\uparrow$ | **83.7**$^\uparrow$ | **86.0**$^\uparrow$ | **87.7**$^\uparrow$ | **83.0**$^\uparrow$ | **85.3**$^\uparrow$ |
| **SIGHAN14** | REALISE (Xu et al., 2021) | 67.8 | 71.5 | 69.6 | 66.3 | 70.0 | 68.1 |
| | Two-Ways (Li et al., 2021a) | - | - | 70.4 | - | - | 68.6 |
| | MLM-phonetics(Zhang et al., 2021a) | 66.2 | **73.8** | 69.8 | 64.2 | **73.8** | 68.7 |
| | LEAD (Li et al., 2022e) | **70.7** | 71.0 | 70.8 | **69.3** | 69.6 | 69.5 |
| | SoftMasked-BERT (Zhang et al., 2020)† | 61.4 | 67.5 | 64.3 | 59.8 | 65.8 | 62.6 |
| | SoftMasked-BERT + DR-CSC | 61.4 | 67.9$^\uparrow$ | 64.5$^\uparrow$ | 60.3$^\uparrow$ | 66.7$^\uparrow$ | 63.4$^\uparrow$ |
| | MacBERT (Cui et al., 2020)† | 63.9 | 65.6 | 64.7 | 61.0 | 62.7 | 61.9 |
| | MacBERT + DR-CSC | 65.6$^\uparrow$ | 68.8$^\uparrow$ | 67.2$^\uparrow$ | 64.8$^\uparrow$ | 68.1$^\uparrow$ | 66.4$^\uparrow$ |
| | SCOPE (Li et al., 2022b) | 70.1 | 73.1 | 71.6 | 68.6 | 71.5 | 70.1 |
| | SCOPE + DR-CSC | 70.2$^\uparrow$ | 73.3$^\uparrow$ | **71.7**$^\uparrow$ | 69.3$^\uparrow$ | 72.3$^\uparrow$ | **70.7**$^\uparrow$ |
| **SIGHAN15** | MLM-phonetics(Zhang et al., 2021a) | 77.5 | 83.1 | 80.2 | 74.9 | 80.2 | 77.5 |
| | REALISE (Xu et al., 2021) | 77.3 | 81.3 | 79.3 | 75.9 | 79.9 | 77.8 |
| | Two-Ways (Li et al., 2021a) | - | - | 80.0 | - | - | 78.2 |
| | LEAD (Li et al., 2022e) | 79.2 | 82.8 | 80.9 | 77.6 | 81.2 | 79.3 |
| | SoftMasked-BERT (Zhang et al., 2020) | 73.7 | 73.2 | 73.5 | 66.7 | 66.2 | 66.4 |
| | SoftMasked-BERT + DR-CSC | 74.0$^\uparrow$ | 78.8$^\uparrow$ | 76.4$^\uparrow$ | 71.6$^\uparrow$ | 76.2$^\uparrow$ | 73.9$^\uparrow$ |
| | MacBERT (Cui et al., 2020) | 71.9 | 77.9 | 74.8 | 68.0 | 73.6 | 70.7 |
| | MacBERT + DR-CSC | 75.8$^\uparrow$ | 78.3$^\uparrow$ | 77.0$^\uparrow$ | 73.6$^\uparrow$ | 76.1$^\uparrow$ | 74.8$^\uparrow$ |
| | SCOPE (Li et al., 2022b) | 81.1 | 84.3 | 82.7 | 79.2 | **82.3** | 80.7 |
| | SCOPE + DR-CSC | **82.9** | **84.8**$^\uparrow$ | **83.8**$^\uparrow$ | **80.3**$^\uparrow$ | 82.3 | **81.3**$^\uparrow$ |
| | *– with d gt* | *89.8* | *94.1* | *91.9* | *85.7* | *89.8* | *87.7* |
| | *– with d/r gt* | *90.2* | *95.2* | *92.6* | *87.0* | *91.9* | *89.4* |

Table 1: The performance of DR-CSC and all baselines. X + DR-CSC means that we combine DR-CSC with model X. "-with d gt" means with detection ground truth during the inference stage and "with d/r gt" means with detection and reasoning ground truth during the inference stage. $\uparrow$ means an improvement compared to the baseline model. Results marked with "†" are obtained by running released codes from corresponding papers.

these three github repositories[4,5,6]. In our experiments, we initialize the weights of SoftMasked-BERT + DR-CSC using the weights of Chinese BERT-wwm (Cui et al., 2021) and we initialize the weights of MacBERT + DR-CSC using the weights of MacBERT (Cui et al., 2020). And the initial weights of SCOPE + DR-CSC are from the Further Pretrained Model proposed by SCOPE (Li et al., 2022b). We set the maximum length of the sentence to 192 which can contain all training samples' length. We train the model with

AdamW optimizer and set the learning rate to $5 \times 10^{-5}$. We set the $\alpha, \beta, \gamma$ all to 1. The optimal models on SIGHAN13/14/15 are obtained by training with batch sizes of 96/96/64 for 20/30/30 epochs, respectively. The best-performing models on SIGHAN13/14/15 were trained using batch sizes of 96/96/64, respectively, for 20/30/30 epochs each. All experiments are conducted on one GeForce RTX 3090.

### 4.2 Experimental Results

From Table 1, we can observe that through the optimization of the module DR-CSC, SoftMasked-BERT, MacBERT, and SCOPE all obtain further

[4]https://github.com/gitabtion/SoftMaskedBert-PyTorch
[5]https://github.com/shibing624/pycorrector
[6]https://github.com/jiahaozhenbang/SCOPE

| Model | Detection | | | Correction | | |
|---|---|---|---|---|---|---|
| | **P** | **R** | **F** | **P** | **R** | **F** |
| SoftMasked-BERT | 73.7 | 73.2 | 73.5 | 66.7 | 66.2 | 66.4 |
| SoftMasked-BERT + DR-CSC | 74.0 | 78.7 | 76.3 | 71.6 | 76.2 | 73.9 |
| – with d/r results from MacBERT + DR-CSC | 74.4 | 78.8 | 76.5 | 72.1 | 76.4 | 74.2 |
| – with d/r results from SCOPE + DR-CSC | 73.3 | 78.5 | 75.8 | 71.6 | 76.6 | 74.0 |
| MacBERT | 71.9 | 77.9 | 74.8 | 68.0 | 73.6 | 70.7 |
| MacBERT + DR-CSC | 75.8 | 78.3 | 77.0 | 73.6 | 76.1 | 74.8 |
| – with d/r results from SoftMasked-BERT + DR-CSC | 75.2 | 79.4 | 77.2 | 75.2 | 76.8 | 74.7 |
| – with d/r results from SCOPE + DR-CSC | 74.7 | 79.0 | 76.8 | 72.8 | 77.0 | 74.8 |
| SCOPE | 81.1 | 84.3 | 82.7 | 79.2 | 82.3 | 80.7 |
| SCOPE + DR-CSC | 82.9 | 84.8 | 83.8 | 80.3 | 82.3 | 81.3 |
| – with d/r results from SoftMasked-BERT + DR-CSC | 82.2 | 83.9 | 83.1 | 80.1 | 81.7 | 80.9 |
| – with d/r results from MacBERT + DR-CSC | 82.2 | 83.7 | 83.0 | 80.2 | 81.7 | 81.0 |

Table 2: The plug-and-play performance analysis on the SIGHAN15 test set. The "d/r results from X" means the detection and reasoning tasks' results come from the X model.

improvements on all test sets and most evaluation metrics, which verifies the effectiveness of the proposed method. Specifically, at correction-level, SCOPE + DR-CSC exceeds SCOPE by 1.0% F1 on SIGHAN13, 0.6% F1 on SIGHAN14, and 0.6% on SIGHAN15. SoftMasked-BERT + DR-CSC exceeds SoftMasked-BERT by 4.0% F1 on SIGHAN13, 0.8% F1 on SIGHAN14, and 7.5% on SIGHAN15. MacBERT + DR-CSC exceeds MacBERT by 4.8% F1 on SIGHAN13, 4.5% F1 on SIGHAN14, and 4.1% on SIGHAN15.

Particularly, SCOPE + DR-CSC achieves new state-of-the-art performance on the three SIGHAN datasets, which reflects the competitiveness of the proposed module.

### 4.3 Analysis and Discussion

#### 4.3.1 Effect of Decomposing the CSC Models

The motivation of our work is to decompose the CSC task into three subtasks to introduce external knowledge and enhance the correction ability of CSC models. Specifically, we divide the CSC task into **Detection**, **Reasoning**, and **Searching** three subtasks, and we believe that the information obtained by detection and reasoning tasks is helpful for learning searching task. To verify our motivation, we feed the detection ground truth into SCOPE + DR-CSC to observe performance changes.

From the results of *"SCOPE + DR-CSC with d gt"* and *"SCOPE + DR-CSC with d/r gt"* in Table 1, we know that the wrong character position information in the detection task and the misspelling type information in the reasoning task are both very helpful to improve the final correction performance. In par-

ticular, when feeding the detection and reasoning tasks' ground truth simultaneously, the correction F-1 of our model reaches 89.4, which shows that the decomposing method is indeed intuitive and natural. Besides, this phenomenon shows the great potential of the DR-CSC module, because the methods we design for detection and reasoning tasks are relatively simple (i.e., naive binary classification).

| Subtask of SCOPE + DR-CSC | Precision | Recall | F1 |
|---|---|---|---|
| Detection | 82.0 | 85.5 | 83.7 |
| Reasoning | 80.5 | 82.8 | 81.6 |
| Searching | 80.3 | 82.3 | 81.3 |

Table 3: The performance of each level of subtasks on the SIGHAN15 test set.

#### 4.3.2 Analysis of Each Subtask

The three subtasks in the CSC models address specific questions: "Which?", "Why?", and "How?". These subtasks are designed to gradually increase in difficulty, progressing from easy to hard. In order to validate the progressive difficulty of our subtasks, we also investigate the individual contributions of each subtask. From Table 3, we reveal a decreasing trend in the performance of the three subtasks: detection, reasoning, and searching. This finding aligns with our initial design intuition and motivation, which suggests that these subtasks follow a progressive difficulty pattern, with each subsequent task becoming more challenging than the previous one.

#### 4.3.3 Plug-and-Play Performance Analysis

We conduct an experiment to assess the compatibility of the proposed DR-CSC module with existing

non-autoregressive CSC models for potential enhancements. This experiment involves utilizing the results of detection and reasoning tasks from various base models. We construct a new search matrix using these results and feed it into the searching task. This approach not only reduces computation costs but also directly enhances the original model. The objective is to determine if the DR-CSC module can effectively integrate with existing models and yield improvements in performance. From Table 2, we observe that each setting performs better than the baseline models on the SIGHAN15 test set. For the Soft-Masked BERT model, utilizing detection and reasoning outputs from MacBERT leads to improved performance compared to training the detection and reasoning tasks on Soft-Masked BERT. For the MacBERT model and SCOPE model, utilizing detection and reasoning output trained on themselves yields the best performance. Overall, the best performance is achieved by utilizing the detection and reasoning task results from SCOPE, possibly due to its different encoders. SCOPE selects ChineseBERT (Sun et al., 2021) as their encoder, which incorporates both the glyph and pinyin information of Chinese characters into language model pretraining. With the assistance of glyph and pinyin information, SCOPE achieves better performance. This approach significantly reduces the time required for retraining a DR module. It allows for the direct utilization of a trained DR module, resulting in substantial resource savings.

| Item | Number |
|---|---|
| Predicted phonological error | 300 |
| Predicted morphological error | 469 |
| Not in phonological confusion set | 0 |
| Not in morphological confusion set | 4 |
| Correct character is filtered out in phonological | 4 |
| Correct character is filtered out in morphological | 7 |

Table 4: We count the outputs of detection and reasoning tasks on the SIGHAN15 test set.

### 4.3.4 Interpretability Analysis

In the searching task of the proposed module, we construct a search matrix that effectively narrows the search space of candidate characters. This optimization greatly enhances the performance of CSC models. Additionally, the CSC models consist of three subtasks that address the questions "Which?Why?How?". This approach enhances interpretability and offers greater potential for understanding.

| Case 1: | |
|---|---|
| **Input Wrong Sentence:** | 刚开始我也收(shōu)不撩(liāo)这样。
I can't accept and agitate it at first either. |
| **SCOPE:** | 刚开始我也收(shōu)不了(liǎo)这样。
I can't accept it at first either. |
| **SCOPE + DR-CSC(w/o Searching):** | 刚开始我也做(zuò)不了(liǎo)这样。
I can't do it at first either. |
| **SCOPE + DR-CSC:** | 刚开始我也受(shòu)不了(liǎo)这样。
I can't stand it at first either. |
| Case 2: | |
| **Input Wrong Sentence:** | 没想到跳着跳着就照(zhào)过半夜了。
I didn't expect to dance and dance light the midnight. |
| **SCOPE:** | 没想到跳着跳着就照(zhào)过半夜了。
I didn't expect to dance and dance light the midnight. |
| **SCOPE + DR-CSC(w/o Searching):** | 没想到跳着跳着就再(zài)过半夜了。
I didn't expect to dance and dance again the midnight. |
| **SCOPE + DR-CSC:** | 没想到跳着跳着就超(chāo)过半夜了。
I didn't expect to dance and dance through the midnight. |

Table 5: Cases from the SIGHAN15 test set show the searching subtask can narrow the search space to enhance the correction ability of the CSC model. We mark the wrong/correct characters.

derstanding. According to the data presented in Table 4, phonological errors account for 39% of the predicted error characters, while morphological errors make up the remaining 61%. Among the morphological error characters, only four are not found in the morphological confusion set, prompting us to assign a search matrix value of one to these characters. Furthermore, out of a total of 769 predicted error characters, only 11 (1.4%) fail to find the correct character within their search matrix. The majority of error characters identified by our module were found to have their correct counterparts within their respective confusion sets. This finding provides a compelling explanation for the performance gains observed in our study.

### 4.3.5 Case Study

From Table 5, we know that SCOPE cannot detect "收", while SCOPE + DR-CSC (w/o Searching) succeeds (even if its correct answer is wrong). This phenomenon indicates the effectiveness of the detection task in DR-CSC, which guides the model to focus on finding the wrong characters at the beginning. In addition, the character (i.e., "收(shōu)") that SCOPE + DR-CSC (w/o Searching) cannot correct is accurately solved by SCOPE + DR-CSC. This is benefited from our searching task, which uses the information provided by detection and

reasoning tasks to construct a fine-grained search matrix so that the model search in the phonological confusion set of "收(shōu)", while avoiding predicting "做(zuò)" whose phonetic is not similar to "收(shōu)". In Case 2, we know that SCOPE cannot detect "照", while SCOPE + DR-CSC (w/o Searching) succeeds (even if its correct answer is wrong). However, SCOPE + DR-CSC can correct it accurately, which benefited from the search matrix provided in the searching task.

## 5 Conclusion

In this paper, we present an enhanced approach for the CSC model by decomposing it into three subtasks. The main objective is to integrate external knowledge and offer improved interpretability. We introduce DR-CSC, a frustratingly easy plug-and-play detection-and-reasoning module that consists of three subtasks with progressive difficulty. By conducting thorough experiments and detailed analyses, we have empirically demonstrated the effectiveness of our decomposition approach, as well as the valuable information provided by each subtask. Furthermore, we believe that DR-CSC holds untapped potential for further exploration.

## 6 Limitations

Our proposed module has three potential limitations that should be acknowledged. Firstly, the detection and reasoning subtasks within our module are relatively straightforward and offer room for improvement. Future research could focus on enhancing the complexity and sophistication of these subtasks to further enhance their performance.

Secondly, our module only considers two types of misspellings: phonological errors and morphological errors. While these two types cover a significant portion of common misspellings, other types of errors may exist that are not addressed in our current framework. Exploring and incorporating additional error types could contribute to a more comprehensive and robust module.

Thirdly, the construction of the search matrix in the searching subtask introduces an additional computational cost during both the training and inference stages. This demands careful consideration in terms of resource allocation and efficiency. Future work should focus on optimizing the search matrix construction process to mitigate the computational overhead while maintaining performance.

Acknowledging these limitations, further advancements can be made to enhance the detection and reasoning subtasks, expand the scope of error types considered, and optimize the computational demands associated with the search matrix construction in the searching subtask.

## Acknowledgements

This research is supported by National Natural Science Foundation of China (Grant No.62276154), Research Center for Computer Network(Shenzhen)Ministry of Education, the Natural Science Foundation of Guangdong Province (Grant No. 2023A1515012914), Basic Research Fund of Shenzhen City (Grant No. JCYJ20210324120012033 and JSGG20210802154402007), the Major Key Project of PCL for Experiments and Applications (PCL2021A06), and Overseas Cooperation Research Fund of Tsinghua Shenzhen International Graduate School (HW2021008).

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
