# OpenReview forum: "A Frustratingly Easy Plug-and-Play Detection-and-Reasoning Module for Chinese Spelling Check"
_EMNLP/2023/Conference — EMNLP 2023 Findings_

### Official Review · Reviewer_frro · 2023-08-04

**Typos Grammar Style And Presentation Improvements:** see reasons to reject
**Soundness:** 3

**Excitement:**

3: Ambivalent: It has merits (e.g., it reports state-of-the-art results, the idea is nice), but there are key weaknesses (e.g., it describes incremental work), and it can significantly benefit from another round of revision. However, I won't object to accepting it if my co-reviewers champion it.

**Paper Topic And Main Contributions:**

This paper proposes a plug-and-play detection-and-reasoning module for Chinese spelling check. The authors decompose the CSC workflow into three subtasks: detection, reasoning, and searching, to leverage external knowledge and enhance the performance of existing non-autoregressive CSC models. They conduct extensive experiments and demonstrate the effectiveness and competitiveness of the proposed modules.

**Questions For The Authors:**

see reasons to reject

**Reasons To Accept:**

1) The paper introduces a novel approach of decomposing the CSC task into three subtasks, which allows for the incorporation of external knowledge and enhances interpretability.

2) The proposed DR-CSC module is compatible with existing non-autoregressive CSC models, making it a practical and easy-to-use tool for enhancing the performance of these models.

3) The paper provides detailed analyses and experimental results to support the effectiveness of the proposed module.

**Reasons To Reject:**

1) The purpose of Figure 1 is not clear, as it fails to effectively communicate the necessary information upon a preliminary glance. The roles of the various icons and colors used in the figure are not explained, leading to confusion about the nature of the CSC task.

2) A similar issue arises with Figure 2. It remains unclear what each color signifies and it is not explicitly mentioned which parts were proposed by the user and which parts are tunable parameters.

3) There seems to be an incorrect portrayal of the "multi-task" aspect. While the authors assert that their approach benefits from multi-task learning in line 295, the paper seems to solely focus on the CSC task.

4) The authors purport their method to be "plug-and-play" and to incorporate external language more naturally. However, their experiments are based on previously proposed CSC models such as SoftMasked-BERT, MacBERT, and SCOPE, which, I believe, are already equipped to handle CSC tasks. I question whether the proposed plug-and-play module can function effectively with pure PLMs, thus making the plug-and-play feature more substantial.


**Reproducibility:**

4: Could mostly reproduce the results, but there may be some variation because of sample variance or minor variations in their interpretation of the protocol or method.

**Reviewer Confidence:**

3: Pretty sure, but there's a chance I missed something. Although I have a good feel for this area in general, I did not carefully check the paper's details, e.g., the math, experimental design, or novelty.

---

> ### Author Rebuttal · Authors · 2023-08-28
>
> Thanks for your appreciation of our work's contributions. We are honored to have the opportunity to discuss with you.
>
>
> ---
>
>
> **Q1**: The purpose of Figure 1 is not clear, as it fails to effectively communicate the necessary information at a preliminary glance. The roles of the various icons and colors used in the figure are not explained, leading to confusion about the nature of the CSC task.
> **A1**: Thank you for your feedback, we will make the necessary revisions to the figure and provide explicit explanations for each element.
>
>
> ---
>
>
> **Q2**: A similar issue arises with Figure 2. It remains unclear what each color signifies and it is not explicitly mentioned which parts were proposed by the user and which parts are tunable parameters.
> **A2**: We appreciate your observation and comment regarding Figure 2. Your feedback highlights an important aspect that needs clarification. To address this issue, we will thoroughly revise the captions associated with the model diagram.
>
>
> ---
>
>
> **Q3**: There seems to be an incorrect portrayal of the "multi-task" aspect. While the authors assert that their approach benefits from multi-task learning in line 295, the paper seems to solely focus on the CSC task.
> **A3**: Thank you for raising this important point.  To address this issue, we want to clarify that we actually decomposed the CSC task into three distinct subtasks. These subtasks were then jointly learned using a multi-task learning approach. Regrettably, we acknowledge that our presentation didn't emphasize this aspect adequately, leading to the misunderstanding that our focus was solely on the CSC task. We will revise our manuscript to provide a more explicit and detailed explanation.
>
>
> ---
>
>
> **Q4**: The authors purport their method to be "plug-and-play" and to incorporate external language more naturally. However, their experiments are based on previously proposed CSC models such as SoftMasked-BERT, MacBERT, and SCOPE, which, I believe, are already equipped to handle CSC tasks. I question whether the proposed plug-and-play module can function effectively with pure PLMs, thus making the plug-and-play feature more substantial.
> **A4**: We appreciate your thoughtful inquiry into the "plug-and-play" aspect of our approach. The rationale behind selecting three distinct baseline models—SoftMasked-BERT, MacBERT, and SCOPE—was to demonstrate the adaptability and effectiveness of our proposed DR-CSC module in a plug-and-play fashion.
> SoftMasked-BERT is a classic CSC model, **and MacBERT is a pure PLM model with a structure like BERT**. SCOPE is the current state-of-the-art CSC model. By integrating our DR-CSC module with these diverse baselines, we aimed to establish the module's seamless integration and effectiveness across various existing approaches.
> MacBERT is a PLM proposed by Cui et al. In the CSC field, MacBERT is treated as a pure PLM baseline similar to BERT but with better performance [1].
>
>
> ---
>
>
> **Reference**
> [1] Contextual similarity is more valuable than character similarity: An empirical study for chinese spell checking. ICASSP 2023.

---

### Official Review · Reviewer_cyLs · 2023-08-04

**Soundness:** 2

**Excitement:**

2: Mediocre: This paper makes marginal contributions (vs non-contemporaneous work), so I would rather not see it in the conference.

**Paper Topic And Main Contributions:**

The paper designs the DR-CSC module, a mechanism that guides the model to correct Chinese spelling errors by integrating a confusion set. The effectiveness of this method has been confirmed by experimental results.



**Questions For The Authors:**

1. The method proposed in this paper uses prior knowledge from the confusion set. Would there be a more pronounced improvement in a low-resource scenario?

2. I would be interested to hear the authors' plan to apply the method from this paper to a decoder-only language model (LLM).

**Reasons To Accept:**

1. The design of the three subtasks improves interpretability.

2. The method section of the paper is well-articulated and clear.

**Reasons To Reject:**

1. The information from the confusion set could potentially be directly incorporated into an end-to-end model for learning. This pipeline approach may lead to error accumulation, thereby reducing the final performance.

2. The improvements observed from the experiment were not significant, and there's a lack of multiple experiments with averaged results and variance analysis.

3. The scope of errors covered is limited to only two types, phonological errors and morphological errors. Problems outside of these errors may not be addressed.

4. The analysis of the three subtasks is too simplistic. Further analysis is needed to understand the reasons for improvements and potential areas for enhancements.

**Reproducibility:**

3: Could reproduce the results with some difficulty. The settings of parameters are underspecified or subjectively determined; the training/evaluation data are not widely available.

**Reviewer Confidence:**

4: Quite sure. I tried to check the important points carefully. It's unlikely, though conceivable, that I missed something that should affect my ratings.

---

> ### Author Rebuttal · Authors · 2023-08-28
>
> We are grateful for your positive comments and insightful reviews. Please find below our point-by-point responses:
>
>
> ---
>
>
> **Q1**: The information from the confusion set could potentially be directly incorporated into an end-to-end model for learning. This pipeline approach may lead to error accumulation, thereby reducing the final performance.
> **A1**: Thank you very much for your valuable feedback. The improvements brought about by DR-CSC directly come from the utilization of a search matrix constructed using the confusion set in the Searching subtask. If the search matrix does not contain the correct corresponding Chinese character, the model will not be able to correct that error. The reasons for such error accumulation could be twofold: Firstly, the Detection subtask might incorrectly identify characters that are not actually errors as typos; secondly, there might be misclassifications in the Reasoning subtask. These factors contribute to error accumulation, ultimately leading to performance degradation.
> We have taken steps to mitigate the effects of both factors. Firstly, **during the construction of the confusion set, we ensured that the corresponding characters were included.** This approach helps alleviate errors caused by the misidentification of non-erroneous characters as errors. Furthermore, **the sets of phonetically similar characters and visually similar characters are not mutually exclusive**; they exhibit a certain degree of intersection. This characteristic further mitigates the impact of error accumulation.
> Finally, based on our experimental statistics as presented in "4.3.4 Interpretability Analysis", **the proportion of correct characters not present in their corresponding confusion sets is remarkably low only accounting for 1.4%**. This indicates that the performance loss resulting from **error accumulation fails to overshadow the gains our method introduces.** It can be viewed as a trade-off that underscores the effectiveness of our approach.
>
>
> ---
>
>
> **Q2**: The improvements observed from the experiment were not significant, and there's a lack of multiple experiments with averaged results and variance analysis.
> **A2**:  **We need to clarify that the experimental results presented in the table are the averages of five individual experiments.** However, we acknowledge the need for a more comprehensive approach, involving multiple experiments and a thorough variance analysis to ensure the robustness of our findings. Your feedback highlights an essential aspect of experimental validation, and we will certainly incorporate this suggestion into our future research.
>
>
> ---
>
>
> **Q3**: The scope of errors covered is limited to only two types, phonological errors and morphological errors. Problems outside of these errors may not be addressed.
> **A3**: In the field of CSC, **the primary errors are primarily attributed to phonological and morphological errors, aligning with the inherent nature of Chinese characters.** The reason for not incorporating a third error type can be explained by the following factors:
> - **Low Incidence**: Other error types comprise a minimal proportion of errors, as indicated by our analysis of the error types in the SIGHAN15 dataset, the "other error types" only account for 1.9%.  Given this low incidence, allocating a new error type for these cases would likely decrease efficiency. Additionally, setting up a new error type would necessitate constructing new confusion sets, which might not be proportional in terms of cost and benefit.
> - **Coverage by Existing Sets**: The current phonetical and visual confusion sets to some extent encompass some characters that could be classified as "other errors."
>
> Lastly, according to our experimental statistics, the proportion of correct characters not present in their corresponding confusion sets is extremely low. This implies that the absence of certain error types in our confusion sets does not necessarily imply an inability of our model to address those errors.
>
>
> ---
>
>
> **Q4**: The analysis of the three subtasks is too simplistic. Further analysis is needed to understand the reasons for improvements and potential areas for enhancements.
> **A4**: **Regarding the analysis of the three subtasks, we conducted three distinct experiments**. First, we input ground truth labels for the Detection and Reasoning subtasks into the model for error correction during the inference. The results are summarized in the table below:
> | Model | D-P | D-R | D-F | C-P | C-R | C-F |
> |------------|-------------------|-------------------|--------------|-------------------|-------------------|--------------|
> | SCOPE + DR-CSC   | 82.9    | 84.8   | 83.8   |80.3| 82.3 |81.3|
> | -with d gt     | 89.8 | 94.1 |91.9 |85.7 |89.8 |87.7|
> |-with d/r gt    | 90.2 | 95.2 |92.6 |87.0 |91.9 |89.4|
>
> This experimental setup aimed to demonstrate the effectiveness of decomposing the CSC task into three subtasks, highlighting the potential of DR-CSC. Moreover, detailed performance metrics for all three subtasks are provided in Table 3 of the paper.
> In addition, we conducted a quality assessment of the search matrix in the search task. The evaluation revealed that only 1.4% of the characters were unable to find the correct character within their corresponding confusion sets, indicating minimal error accumulation. Furthermore, the Case Study reaffirms the effectiveness of the proposed modules.
> To address your concerns about the simplicity of the analysis, we acknowledge the need for a more comprehensive investigation into the factors contributing to the observed improvements and areas where enhancements can be made. We plan to delve deeper into these aspects by considering various influencing factors and conducting additional experiments to provide a more insightful analysis.
>
>
> ---
>
>
> **Q5**: The method proposed in this paper uses prior knowledge from the confusion set. Would there be a more pronounced improvement in a low-resource scenario?
> **A5**: We sincerely appreciate the insights you have provided, as they are instrumental in improving and elevating the quality of our research. In the field of CSC, several mainstream approaches are currently being explored:
> - **Introduction of Multimodality**: Some methods integrate multimodal information [1].
> - **Variations of Pretraining Techniques**: Another direction involves adapting variations of pretraining methods [2,3].
> - **Efficient Utilization of External Knowledge**: External knowledge, such as confusion sets and dictionary information, plays a substantial role in enhancing CSC [4,5].
>
> Our approach specifically focuses on efficiently incorporating external knowledge. Leveraging this external knowledge has consistently been a means to enhance the effectiveness of CSC methods.
> Hence, in a low-resource scenario, where access to extensive data might be limited, the utilization of external knowledge, as demonstrated in our approach, could indeed yield more significant improvements. The knowledge encoded within **the confusion sets becomes even more valuable in situations where the model has less data to learn from.**
>
>
> ---
>
>
> **Q6**: I would be interested to hear the authors' plan to apply the method from this paper to a decoder-only language model (LLM).
> **A6**: Thank you for highlighting this point. Indeed, the potential application of our method to a decoder-only language model (LLM) presents an intriguing avenue for future research. **Injecting the knowledge derived from our approach into an LLM is a promising direction that aligns well with our ongoing efforts.** Exploring how to effectively incorporate our technique into a decoder-based framework and leverage its benefits within that context will be a focus of our forthcoming work. Your inquiry underscores the exciting prospects for extending the utility of our approach to different language modeling paradigms.
>
>
> ---
>
>
> **References**
> [1] Read, Listen, and See: Leveraging Multimodal Information Helps Chinese Spell Checking. ACL-IJCNLP 2021.
> [2] The Past Mistake is the Future Wisdom: Error-driven Contrastive Probability Optimization for Chinese Spell Checking. ACL 2022.
> [3] Improving Chinese Spelling Check by Character Pronunciation Prediction: The Effects of Adaptivity and Granularity. EMNLP 2022.
> [4] Learning from the Dictionary: Heterogeneous Knowledge Guided Fine-tuning for Chinese Spell Checking. EMNLP 2022.
> [5] SpellGCN: Incorporating Phonological and Visual Similarities into Language Models for Chinese Spelling Check. ACL 2020.

---

### Official Review · Reviewer_iyvk · 2023-08-11

**Soundness:** 3
**Typos Grammar Style And Presentation Improvements:** Line 538 6 Limitations -> Limitations

**Excitement:**

4: Strong: This paper deepens the understanding of some phenomenon or lowers the barriers to an existing research direction.

**Paper Topic And Main Contributions:**

This paper proposes a simple but effective plug-and-play module named DR-CSC (Detection-and-Reasoning module for Chinese Spelling Check). They decompose CSC task into three sub-tasks (Detection, Reasoning and Search) in a human-intuitive way. The overall structure of the module seems to be a PLM encoder with three prediction heads, which model the three sub-tasks of retrieval, reasoning and search respectively, and adopt the paradigm of multi-task learning to update parameters. This paper conducts extensive experimental analysis on SIGHAN13/14/15 datasets. The evaluation results on show that DR-CSC's SOTA performance and good plug-and-play ability.


**Questions For The Authors:**

Question A: Although previous work has been using SIGHAN13/14/15 datasets, I am concerned that they may be somewhat outdated. Have you considered experimenting with some newer datasets (such [1] or [2] )?

Quetion B: I don't understand a little about "-with d gt" and "with d/r gt" in Table 1. Do they mean to using a method similar to teach-forcing during training or to input ground truth during testing?

Question C: In my opinion the form of pipline seems to be more intuitive to decompose CSC task. Can you provide some reasons or result analysis for using multi-task learning instead of pipline?


References

[1]https://github.com/nghuyong/cscd-ime

[2]https://github.com/Aopolin-Lv/ECSpell

**Reasons To Accept:**

1. The method is simple but effective. And the proposed model shows good potential in incorporating new expert knowledge.

2. The experimental analysis is sufficient and convincing.

3. Clear and direct structure of paper writing.

**Reasons To Reject:**

1. The improvement is low.  While the module does consistently improve CSC results compared to all prior approaches, it is not clear that the magnitude of the improvement is worth the cost, especially with double PLM parameters.

2. Experiments use fewer datasets. See "Questions For The Authors" for details.

**Reproducibility:**

4: Could mostly reproduce the results, but there may be some variation because of sample variance or minor variations in their interpretation of the protocol or method.

**Reviewer Confidence:**

3: Pretty sure, but there's a chance I missed something. Although I have a good feel for this area in general, I did not carefully check the paper's details, e.g., the math, experimental design, or novelty.

---

> ### Author Rebuttal · Authors · 2023-08-28
>
> Thanks for your valuable comments. We attach great importance to your questions and hope to address your concerns.
>
> ---
>
>
> **Q1**: The improvement is low. While the module does consistently improve CSC results compared to all prior approaches, it is not clear that the magnitude of the improvement is worth the cost, especially with double PLM parameters.
> **A1**: Our module can be integrated with any non-autoregressive CSC model. By incorporating our module and retraining, improvements over the baseline model's performance can be achieved. The costs of the model include various aspects, such as the size of the model's parameters and the computational resources required for training. Furthermore, the pre-trained DR module can be directly combined with a new CSC model, leading to an enhancement in the new model's performance. This scenario would entail the utilization of double PLM parameters. **However, it's important to note that there is no need to retrain a new DR module in this case, resulting in substantial cost savings. This situation presents a trade-off.**
>
>
> ---
>
>
> **Q2**: Although previous work has been using SIGHAN13/14/15 datasets, I am concerned that they may be somewhat outdated. Have you considered experimenting with some newer datasets (such as [1] or [2] )?
> **A2**: Absolutely, thank you for bringing this up. We are aware of the CSCD-IME dataset, which is an open-source collection specifically errors caused by the pinyin input method.
> Furthermore, ECSpell offers a diverse set of domains for CSC, incorporating data from legal, medical, and official document contexts.
> We believe that the datasets mentioned above are all too domain-relevant and contrary to our goal of
> **However, we think that the datasets mentioned above are all too domain-relevant and deviate from our goal of achieving a more universally applicable approach to CSC.** In this context, the SIGHAN datasets excel due to their broader, more general nature.
> It's noteworthy that as of the latest ACL 2023 [1-4] and EMNLP 2022 [5-8] conferences, **these datasets have not been widely adopted.** However, we greatly appreciate your suggestion, and we plan to conduct more insightful research on these datasets in the future.
>
>
> ---
>
>
> **Q3**: I don't understand a little about "-with d gt" and "with d/r gt" in Table 1. Do they mean to use a method similar to teach-forcing during training or to input ground truth during testing?
> **A3**: They actually refer to the latter one – **input ground truth during testing**. We understand that the phrasing in the caption of Table 1 may need improvement to better reflect this.
>
>
> ---
>
>
> **Q4**: In my opinion, the form of pipeline seems to be more intuitive to decompose CSC task. Can you provide some reasons or result analysis for using multi-task learning instead of pipeline?
> **A4**: While the pipeline approach indeed has its merits, it's important to consider the training cost associated with sequentially training these modules. **Each step in the pipeline necessitates independent training, potentially leading to a lengthier and more resource-intensive training process.** In contrast, we made the decision to adopt a multi-task learning framework, allowing the model to simultaneously learn from multiple tasks. In our experiments, we found that the multi-task learning approach not only streamlined the training process but also demonstrated improved performance on various evaluation metrics. This suggests that the model's ability to learn from multiple tasks simultaneously contributed to its enhanced performance compared to a traditional pipeline approach.
>
>
> ---
>
>
> References
> [1] UMRSpell: Unifying the Detection and Correction Parts of Pre-trained Models towards Chinese Missing, Redundant, and Spelling Correction. ACL 2023.
> [2] Disentangled Phonetic Representation for Chinese Spelling Correction. ACL 2023.
> [3] PTCSpell: Pre-trained Corrector Based on Character Shape and Pinyin for Chinese Spelling Correction. ACL 2023.
> [4] Investigating Glyph-Phonetic Information for Chinese Spell Checking: What Works and What’s Next?. ACL 2023.
> [5] An Error-Guided Correction Model for Chinese Spelling Error Correction. EMNLP 2022.
> [6] Learning from the Dictionary: Heterogeneous Knowledge Guided Fine-tuning for Chinese Spell Checking. EMNLP 2022.
> [7] WSpeller: Robust Word Segmentation for Enhancing Chinese Spelling Check. EMNLP 2022.
> [8] Improving Chinese Spelling Check by Character Pronunciation Prediction: The Effects of Adaptivity and Granularity. EMNLP 2022.

---

### Meta-Review · Area_Chair_vDFX · 2023-09-27

**Recommendation:** 4

**Metareview:**

The reviewers overall agreed that the proposed method is simple and effective and that the experiments are sound, with some concerns regarding the amount of improvement over the baselines. The authors have tried to address most reviewers' questions, by providing a fair amount of discussion and justification.

---

### Decision · Program_Chairs · 2023-10-07

**Decision:**

Accept-Findings

**Comment:**

The reviewers overall agreed that the proposed method is simple and effective and that the experiments are sound, with some concerns regarding the amount of improvement over the baselines. The authors have tried to address most reviewers' questions, by providing a fair amount of discussion and justification.